

# Elevated estradiol levels on hCG trigger day adversely effects on the clinical pregnancy rates of blastocyst embryo transfer but not cleavage-stage embryo transfer in fresh cycles: a retrospective cohort study

Yue Meng[1,*], Linlin Tao[1,*], Tingting Xia[1], Jieru Zhu[1], Xiaoqi Lin[1], Wen Zhou[1], Yuxia Liu[2], Jianping Ou[1] and Weijie Xing[1]

[1] Reproductive Medicine Center, the Third Affiliated Hospital of Sun Yat-Sen University, Guangzhou, Guangdong province, China
[2] Reproductive Medicine Center, the First People's Hospital of Kashi Prefecture, Affiliated Kashi Hospital of Sun Yat-Sen University, Kashi, China
[*] These authors contributed equally to this work.

Corresponding authors
Jianping Ou, oujp3@mail.sysu.edu.cn
Weijie Xing, xing-weij@mail.sysu.edu.cn

## ABSTRACT

**Background.** Elevated estradiol ($E_2$) levels are an inevitable outcome of the controlled ovulation hyperstimulation. However, the effect of this change on pregnancy is still uncertain. Our study aimed to analyze the impact of increased serum $E_2$ at the day of human chorionic gonadotropin (hCG) administration on the clinical outcomes of women with fresh embryo transfer (ET) cycles.

**Methods.** This study included 3,009 fresh ET cycles from October 2015 to September 2021. Based on the stage of embryos transferred, these cycles were categorized into the cleavage group and blastocyst group. Both groups were then divided into four sets according to $E_2$ levels when hCG was administered: set 1 ($E_2 \leq 2,000$ pg/ml), set 2 ($E_2 = 2,001$–$3,000$ pg/ml), set 3 ($E_2 = 3,001$–$4,000$ pg/ml), and set 4 ($E_2 > 4,000$ pg/ml). The primary outcome was the clinical pregnancy rate (CPR). Binary logistics regression analysis was established to explore the association between CPR and E2 levels. Specifically, the threshold effect of serum E2 on CPR was revealed using the two-piecewise linear regression analyses.

**Results.** The multivariate regression model in the cleavage group showed that patients' CPR in set 4 was 1.59 times higher than those in reference set 1, but the statistical difference was insignificant ($P = 0.294$). As for the blastocyst group, patients in set 4 had a lower CPR with adjusted ORs of 0.43 ($P = 0.039$) compared to patients in set 1. The inflection point for the blastocyst group was 39.7 pg/dl according to the results of the two-piecewise linear regression model. When $E_2$ levels were over the point, the CPR decreased by 17% with every 1 pg/dl increases in serum $E_2$ (adjusted OR = 0.83, 95% CI [0.72–0.96], $P = 0.012$).

**Conclusions.** Elevated $E_2$ levels (>39.7 pg/dl) on hCG trigger day were associated with decreased CPR in patients with fresh blastocyst ET. However, it had no similar effect on the CPR of patients with fresh cleavage-stage ET.

## INTRODUCTION

Assisted reproductive technology (ART) has evolved rapidly since its advent and has become the most effective treatment for infertile patients (*De Geyter, 2019*). Advancements in this field include controlled ovulation hyperstimulation (COS) using gonadotropin, which can artificially increase the number of oocytes retrieved (*Arslan et al., 2005*). However, supraphysiologic estradiol ($E_2$) levels resulting from COS may have a detrimental effect on the endometrial receptivity (*Fatemi & Popovic-Todorovic, 2013*; *Ullah et al., 2017*). *Fatemi & Popovic-Todorovic (2013)* reported that a high $E_2$ concentration in fresh embryo transfer (ET) cycles generated early endometrial luteinization and advanced the window of implantation, inducing a reduced embryo implantation rate (*Fatemi & Popovic-Todorovic, 2013*). High $E_2$ levels on the day of human chorionic gonadotropin (hCG) administration are also related to adverse neonatal outcomes, such as decreased clinical pregnancy rate (CPR), increased low birthweight (LBW) and the rate of small for gestational age (SGA) babies (*Hajshafiha et al., 2021*; *Li et al., 2019*; *Liu et al., 2017*). By contrast, a meta-analysis by Glykeria et al. concluded that there is insufficient evidence to support the correlation between $E_2$ levels on the day of hCG administration and the probability of pregnancy (*Karatasiou et al., 2020*). Another large retrospective cohort study of 8,501 patients with ART did not find a relationship between peak serum $E_2$ after COS and neonatal birthweights (*Huang et al., 2020*). Therefore, the effect of elevated $E_2$ levels on ART outcomes remains uncertain.

In fresh ET cycles, the embryo stage is essential to pregnancy outcomes (*Glujovsky et al., 2016*). Hsieh et al. performed a retrospective cohort study comprising 9,090 fresh ET. The results showed a small significant difference in the implantation rate that favored the transfer on day 5 ($P = 0.04$). However, there were no differences between day 3 and day 5 transfers on live birth ($P = 0.27$) and CPR ($P = 0.11$) (*Hsieh et al., 2021*). Multicenter, randomized controlled trials (RCTs) analyzed by *Wei et al. (2019)* suggested that fresh blastocyst transfers showed decreased live birth rates *versus* frozen blastocyst transfers in women with good prognoses. However, two large RCTs with similar experimental design in cleavage-stage ET showed no difference in the rates of implantation, pregnancy, and livebirth (*Shi et al., 2018*; *Vuong et al., 2018*). The mechanism underlying these findings may be the induction of a partial implantation window that closed in response to an increased $E_2$ concentration. COS-induced supraphysiological $E_2$ levels can result in premature endometrium, leading to the unsynchronized development of endometrium and embryo, particularly when transferring blastocysts on the fifth day after ovum pick-up (*Manvelyan et al., 2022*). Based on these data, we presume that the increased $E_2$ levels have different effects on endometrial receptivity between fresh cleavage-stage and blastocyst ET. Accordingly, our study aimed to retrospectively investigate the correlation between serum $E_2$ on the day of hCG administration and CPR according to different embryo stages in fresh cycles.

## MATERIALS & METHODS

### Patient selection

This single-center, retrospective cohort study was performed at the Reproductive Medicine Center of the Third Affiliated Hospital of Sun-Yat Sen University. This study protocol was approved by the Institutional Ethics Committee of the Third Affiliated hospital of Sun Yat-Sen university. All participants had signed the informed consent (Institutional Review Board approval: II 2023-051-01). Patients who underwent fresh ET cycles from October 2015 to September 2021 were included. Figure 1 summarizes the flow chart of this study. A total of 9,841 fresh cycles were initially included. Cases were excluded if they met the following criteria: (i) cycles were canceled due to no oocyte retrieval or no ET for the physical condition; (ii) no top-level embryos were produced (based on embryo stage on the day of transfer, embryos were scored by their morphologic manifestation under the light microscope) (*Alpha Scientists in Reproductive Medicine & ESHRE Special Interest Group of Embryology, 2011*; *Gardner, Lane & Schoolcraft, 2000*) ; (iii) core data was missing (*e.g.*, pregnancy record or $E_2$ level on the hCG trigger day); (iv) endometrial or uterine factors impacted pregnancy, including endometrial polyps, fibroid uterus, and uterine effusion; (v) patients experienced early-onset severe ovarian hyperstimulation syndrome (OHSS) during a fresh ET cycle (the severity grading of OHSS was based on clinical symptoms and laboratory indicators (*Nelson, 2017*). (vi) patients lost to follow-up. After screening, a total of 3,009 cycles were available for this study. According to the embryo stage at transfer day, all fresh ET cycles were then classified into the cleavage group (transfer on day 3) and the blastocyst group (transfer on day 5).

### Controlled stimulation and embryo transfer procedures

Patients received one of two COS protocols: the gonadotropin releasing hormone agonist (GnRH-a) long protocol or the gonadotropin releasing hormone antagonist (GnRH-ant) protocol. The appropriate protocol was selected by the clinician for each patient on the basis of individual characteristics.

The GnRH-a long protocol included long-acting GnRH-a or short-acting GnRH-a, which were different in injection dosage during mid-luteal phases. Patients received one dose of 1.0 mg triptorelin (Gonapeptyl; Ferring, France) in the long-acting GnRH-a protocol. As for short-acting GnRH-a protocol, patients alternatively received a daily injection of 0.1 mg of triptorelin. Pituitary-ovarian suppression was established based on serum LH <5 mIU/ml and $E_2$ <50 pg/ml. Next, a daily gonadotropin dose of 100-300 IU was used for ovarian stimulation and the amount was adjusted according to the ovarian response until the day of hCG administration.

For the GnRH-ant protocol, ovarian stimulation commenced with daily 75-300 IU gonadotropin from days 2 to 3 of menstruation. The clinician assessed and adjusted the dosage of gonadotropin according to ultrasound readings and serum $E_2$ levels. A daily dose of 0.25 mg cetrorelix (Cetrotide; Serono, Geneva, Switzerland) was given when the dominant follicle reached 14 mm and lasted until the day of hCG administration.

When at least two follicles $\geq$ 18 mm diameter were identified, a dose of 4,000 to 10,000 IU hCG (Lizhu, Guangdong, China) was used to trigger the follicle maturation.

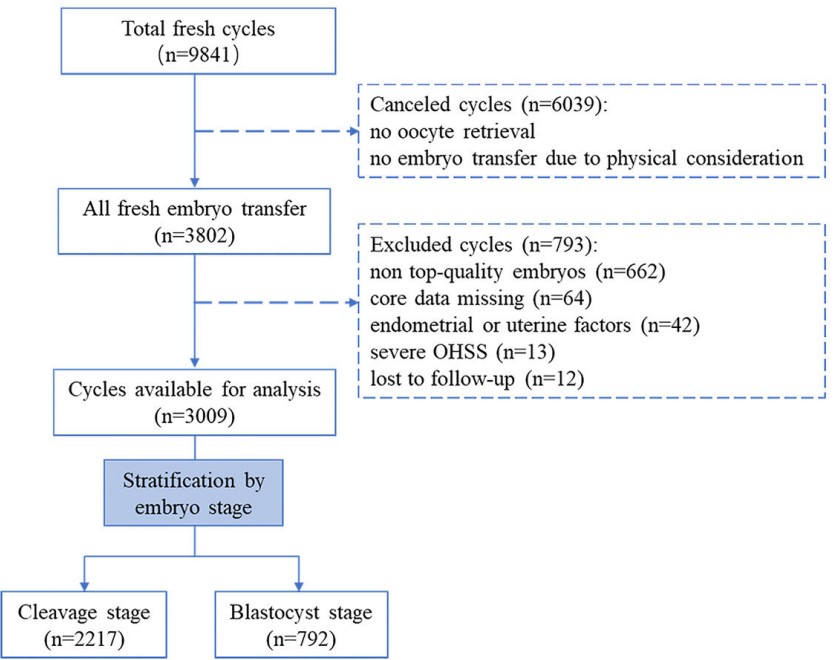

**Figure 1  Flow chart of the study.**

Oocyte retrieval was scheduled at 34–36 h after the injection of hCG under the guidance of a vaginal ultrasound. After retrieval, oocytes were inseminated using the standard fertilization method, in-vitro fertilization (IVF) or intracytoplasmic sperm injection (ICSI), depending on the sperm's quality. The male partner in this study had to be present for semen analysis on at least two separate visits. Normally, semen samples were derived from ejaculated sperm. If the male partner had non-obstructive azoospermia, semen was acquired using testicular or percutaneous epididymal sperm aspiration methods. ICSI was performed if any of the following indications existed: severe oligospermia (sperm density $<5 \times 10^6$ ml in two out of three semen analysis); severe weak spermatozoa (progressive motility ratio <10% in two out of three semen analysis); severe teratospermia (according to the sperm morphological criteria, the ratio of normal morphological sperm was <1% in two out of three semen analysis) (*Esteves, 2022*; *Gatimel et al., 2017*).

The embryos were incubated at 37 °C in a humid gas phase with a 4% oxygen, 6% carbon dioxide, and 90% nitrogen mixture. Normal fertilization is defined as the appearance of two pronuclei and two polar bodies within 16–18 h of insemination (the fertilization rate of embryo in these cycles ranged from 55.6% to 90%). Embryos scoring standard referred to the Society for Assisted Reproductive Technology (SART) scoring system (*Racowsky et al., 2011*). Cleavage-stage embryo was graded according to the number of blastomeres, the percentage of fragments, and the size of the blastomeres. A good quality cleavage-stage embryo was described as having a blastomere with seven to nine cells of an A or B grade. The blastocyst scoring included the stage of blastocyst expansion, the density and number

of cells in the inner cell mass, the regularity as well as the cohesion of the trophectoderm. A blastocyst embryo above 3BB on day 5 was defined as being of good quality.

Fresh ET was planned based on a receptive endometrium state (acceptable morphology; endometrial thickness ≥ 8 mm). Cleavage-stage embryos were transferred on day 3 and blastocysts were transferred on day 5 after ovum pick-up. The luteal phase was supported by daily vaginal or intramuscular progesterone after fresh ET. If the pregnancy was successful, this administration continued until ten weeks of gestation.

## Data collection and outcome measurement

The baseline demographic and clinical characteristics of the study subjects were acquired from our database. Basal sexual hormone levels involved follicle stimulating hormone (FSH, mIU/ml), luteinizing hormone (LH, mIU/ml), and estradiol ($E_2$, pg/ml), detected on the first to third day of menstruation. COS indicators comprised the ovarian stimulation protocol, sexual hormone levels on the hCG trigger day, and the number of retrieved oocytes. Endometrial thickness (mm) on the hCG trigger day was measured by ultrasonography (*Simeonov et al., 2020*). The number of inviable embryos and embryos transferred was also noted.

CPR per fresh ET cycle was the primary measurement of the study. Clinical pregnancy was confirmed by ultrasound to detect gestational sacs. All pregnant cases were followed up to acquire the delivery data. The secondary outcomes included miscarriage rate (spontaneous loss of intrauterine pregnancy before 28 gestational weeks), ectopic pregnancy rate (gestational sac outside of uterine confirmed by ultrasound), live birth rate (the delivery of one or more viable infant), singleton live birth rate, preterm birth rate (delivery before 37 weeks), fetal weight, and the rates of LBW (defined as neonatal weight <2,500 g), and SGA (defines as an infant with a birthweight <10th percentile for its gestational age).

## Statistical analysis

Patients of both embryo stage groups were categorized into four sets according to the serum $E_2$ on the hCG day: $E_2 \leq 2,000$ pg/ml (set 1), $E_2 = 2,001$–$3,000$ pg/ml (set 2), $E_2 = 3,001$–$4,000$ pg/ml (set 3), $E_2 > 4,000$ pg/ml (set 4). The patients of set 1 ($E_2 < 2,000$ pg/ml) were taken as reference. Continuous variables were presented as means ± standard deviation (normally distributed variables) or median with quartile (non-normally distributed variables), and differences among different sets were compared by one-way analysis of variance or Kruskal–Wallis test, respectively. Categorical variables were presented as a number with percentage, and difference were compared by chi-square test.

Binary logistics analysis was established to assess the association between serum $E_2$ on the hCG trigger day and CPR. Firstly, univariate analysis was applied to screened the potential risk factors for CPR. Variables were treated as confounders in the next multivariate regression if they were significantly associated with CPR in univariate analysis, or they changed the matched odds ratios (ORs) by more than 10 percent. The results of the previous study and clinical experience were also considered (*Brodin et al., 2009*; *Kahyaoglu et al., 2015*; *Sermondade et al., 2019*; *Wang et al., 2022a*; *Wang et al., 2022b*). The following were selected as confounding factors: age, BMI, infertility duration, fertilization method, sterility

classification, previous cycle attempts, basal FSH/LH/$E_2$ levels, FSH/LH/progesterone (P) levels on hCG trigger day, endometrial thickness on hCG trigger day, number of retrieved oocytes, the number of inviable embryos, and the number of embryos transferred. We assessed the non-linear relationship between serum $E_2$ and CPR based on the spline smoothing plots. The threshold effect of serum $E_2$ on CPR were evaluated using a two-piecewise linear regression and the inflection point was defined using a recurrence method. Crude and adjusted odds ratio (OR) with 95% confidence intervals (CIs) were calculated to present the results. $P < 0.05$ was considered to be statistically significant. Statistical analyses were performed using EmpowerStats software (X&Y Solution, Inc., Boston, MA, USA; https://www.empowerstats.net/en/).

## RESULTS

### Baseline characteristics

A total of 2,217 and 792 fresh cycles were included in cleavage group and blastocyst group, respectively. Table 1 shows a comparison of the clinical characteristics among the different sets of the cleavage group. The patients with $E_2 \leq 2,000$ pg/ml on the trigger day were older on average ($34.5 \pm 5.6$ years), had a higher BMI (21.64, Q1-Q3 = 20.03–23.73, kg/m2), had made more previous ET attempts ($\geq$ once, 27.73%) as well as a higher proportion in secondary infertility (61.75%), compared to patients in other sets. As for ovarian stimulation, patients with lower $E_2$ levels showed a great chance of using the GnRH-ant protocol for the pituitary down regulation. In addition, patients with elevated $E_2$ levels on hCG trigger day exhibited a higher basal LH level, higher LH and P levels on hCG trigger day, a thicker endometrium, more retrieved oocytes and more transferred embryos. Table 2 lists the characteristics among different sets in the blastocyst group. The differences were statistically significant in age, BMI, basal LH level, LH and P levels on hCG trigger day, the number of retrieved oocytes, and the trends were the same as those in the cleavage group.

### Descriptive analysis of clinical outcomes

The pregnancy and perinatal outcomes of both groups according to different $E_2$ levels are presented in Table 3. We observed the increasing trends in CPR ($P < 0.001$), live birth rate ($P = 0.002$), and singleton live birth rate ($P = 0.035$) from set 1 to set 4 of patients in the cleavage group. For the blastocyst group, CPR expressed a decreasing trend ($P = 0.016$), and patients in set 4 had a significantly lower CPR compared to those in set 1 (38.1% *vs.* 56.2%, $P = 0.007$). Although the rates of preterm birth and LBW showed an increasing trend in both embryo stage groups, the differences among $E_2$ level sets of the two groups were not statistically significant.

### Binary logistics analysis of clinical pregnancy rate

The univariate analyses in both groups are listed in Table 4. In the cleavage group, endometrial thickness on trigger day, retrieved oocytes, inviable embryos, and transferred embryos were each positively correlated with CPR. While older age, more previous cycle attempts, longer infertility duration, higher basal FSH, and FSH on hCG trigger day were

**Table 1 Comparison of baseline characteristics between patients with different $E_2$ levels on the trigger day in cleavage group (N = 2217).**

| Items | Set 1 $E_2 \leq 2000$ pg/ml | Set 2 $E_2 = 2001\text{-}3000$ pg/ml | Set 3 $E_2 = 3001\text{-}4000$ pg/ml | Set 4 $E_2 > 4000$ pg/ml |
|---|---|---|---|---|
| N | 1208 | 584 | 320 | 105 |
| Age (years) | $34.5 \pm 5.6$ | $32.3 \pm 5.3$[a] | $32.2 \pm 4.6$[a] | $31.6 \pm 4.5$[a] |
| BMI (kg/m$^2$), median (Q1-Q3) | 21.64 (20.03-23.73) | 21.23 (19.53-23.06)[a] | 20.93 (19.53-22.90)[a] | 20.83 (19.10-22.89)[a] |
| Duration of infertility (years), median (Q1-Q3) | 3.0 (1.0-5.0) | 2.0 (1.0-4.0) | 3.0 (2.0-5.0) | 3.0 (2.0-5.0) |
| Fertilization method, n (%) | | | | |
| IVF | 1011 (83.7%) | 458 (78.4%) | 261 (81.6%) | 81 (77.1%) |
| ICSI | 197 (16.3%) | 126 (21.6%) | 59 (18.4%) | 24 (22.9%) |
| Previous cycle attempts, n (%) | | | | |
| 0 | 873 (72.2%) | 481 (82.4%)[a] | 266 (83.1%)[a] | 89 (84.8%)[a] |
| 1-2 | 269 (22.2%) | 95 (16.2%)[a] | 50 (15.6%)[a] | 14 (13.3%)[a] |
| ≥3 | 66 (5.4%) | 8 (1.4%)[a] | 4 (1.3%)[a] | 2 (1.90%)[a] |
| Sterility classification, n (%) | | | | |
| primary | 462 (38.3%) | 264 (45.2%)[a] | 144 (45.0%) | 55 (52.4%)[a] |
| secondary | 746 (61.7%) | 320 (54.8%)[a] | 176 (55.0%) | 50 (47.6%)[a] |
| Ovarian stimulation protocol, n (%) | | | | |
| GnRH-antagonist protocol | 830 (68.3%) | 319 (54.6%)[a] | 151 (47.2%)[a] | 36 (34.3%)[ab] |
| GnRH-agonist long protocol | 378 (31.3%) | 265 (45.4)[a] | 169 (52.8%)[a] | 69 (65.7%)[abc] |
| Basal FSH (mIU/ml) | $7.3 \pm 2.6$ | $6.6 \pm 1.9$[a] | $6.5 \pm 1.6$[a] | $6.4 \pm 2.0$[a] |
| Basal LH (mIU/ml) | $4.7 \pm 2.0$ | $5.0 \pm 2.1$[a] | $5.3 \pm 2.0$[a] | $5.0 \pm 2.0$ |
| Basal $E_2$ (pg/ml) | $37.7 \pm 17.1$ | $37.0 \pm 16.8$ | $39.2 \pm 15.9$ | $36.6 \pm 16.1$ |
| hCG day FSH (mIU/ml) | $15.7 \pm 4.8$ | $14.8 \pm 4.7$[a] | $14.6 \pm 4.7$[a] | $13.4 \pm 4.3$[ab] |
| hCG day LH (mIU/ml) | $3.2 \pm 2.5$ | $3.0 \pm 2.3$ | $2.7 \pm 2.2$[a] | $2.7 \pm 1.9$ |
| hCG day $E_2$ (pg/ml) | $1309.3 \pm 428.5$ | $2450.1 \pm 282.7$[a] | $3420.9 \pm 294.2$[ab] | $4344.46 \pm 223.8$[abc] |
| hCG day P (ng/ml) | $0.5 \pm 0.3$ | $0.7 \pm 0.3$[a] | $0.8 \pm 0.3$[ab] | $0.8 \pm 0.3$[ab] |
| Endometrial thickness (mm), median (Q1-Q3) | 10.7 (9.3–12.1) | 11.0 (9.6–12.6)[a] | 11.0 (9.6–12.5) | 10.25 (9.03–12.4) |
| No. of retrieved oocytes, median (Q1-Q3) | 6.0 (4.0–8.0) | 10.0 (8.0–13.0)[a] | 12.0 (10.0–15.0)[ab] | 14.0 (12.0–17.0)[abc] |
| No. of inviable embryos, median (Q1-Q3) | 2.0 (2.0–4.0) | 3.0 (2.0–5.0)[a] | 4.0 (2.0–5.0)[a] | 4.0 (2.0–5.0)[a] |
| No. of embryos transferred, n (%) | | | | |
| 1 | 222 (18.4%) | 59 (10.1%)[a] | 32 (10.0%)[a] | 4 (3.8%)[a] |
| 2 | 986 (81.6%) | 525 (89.9%)[a] | 288 (90.0%)[a] | 101 (96.2%)[a] |

Notes.

Data are presented as mean ± standard deviation unless otherwise indicated.

[a] Compared with set 1, $P < 0.05$.

[b] Compared with set 2, $P < 0.05$.

[c] Compared with set 3, $P < 0.05$.

Abbreviations: $E_2$, estradiol; Q, quartile; BMI, body mass index; IVF, in-vitro fertilization; ICSI, intracytoplasmic sperm injection; FSH, follicle stimulating hormone; LH, luteinizing hormone; hCG, human chorionic gonadotropin; P, progesterone.

significantly associated with decreased CPR (all $P < 0.05$). As for the blastocyst group, the number of embryos transferred was positively associated with CPR, and covariates in terms of age, FSH and P on hCG trigger day were negatively correlated with CPR (all $P < 0.05$). $E_2$ levels on hCG trigger day showed different effects for the outcomes in

**Table 2  Comparison of baseline characteristics between patients with different E2 levels on the trigger day in blastocyst group (N = 792).**

| Items | Set 1 $E_2 \leq 2000$ pg/ml | Set 2 $E_2 = 2001–3000$ pg/ml | Set 3 $E_2 = 3001–4000$ pg/ml | Set 4 $E_2 > 4000$ pg/ml |
|---|---|---|---|---|
| N | 178 | 310 | 241 | 63 |
| Age (years) | 32.44 ± 4.90 | 31.57 ± 4.45 | 31.63 ± 4.39 | 31.39 ± 4.47[a] |
| BMI (kg/m$^2$), median (Q1-Q3) | 21.80 (20.15–24.06) | 20.81 (19.53–23.55)[a] | 20.81 (19.53–23.41)[a] | 20.31(19.19–22.89)[a] |
| Duration of infertility (years), median (Q1-Q3) | 3.00 (1.00–5.00) | 3.00 (2.00–4.00) | 2.00 (1.00–4.00) | 2.00 (1.00–5.00) |
| Fertilization method, n (%) | | | | |
| IVF | 151 (84.8%) | 244 (78.7%) | 190 (78.8%) | 58 (92.1%) |
| ICSI | 27 (15.2%) | 66 (21.3%) | 51 (21.2%) | 5 (7.9%) |
| Previous cycle attempts, n (%) | | | | |
| 0 | 144 (80.9%) | 268 (86.5%) | 201 (83.4%) | 58 (92.1%) |
| 1-2 | 29 (16.3%) | 37 (11.9%) | 38 (15.8%) | 4 (6.4%) |
| ≥3 | 5 (2.8%) | 5 (1.6%) | 2 (0.8%) | 1 (1.5%) |
| Sterility classification, n (%) | | | | |
| primary | 72 (40.4%) | 138 (44.5%) | 109 (45.2%) | 19 (30.2%) |
| secondary | 106 (59.6%) | 172 (55.5%) | 132 (54.8%) | 44 (69.8%) |
| Ovarian stimulation protocol, n (%) | | | | |
| GnRH-antagonist protocol | 119 (66.9%) | 207 (66.8%) | 144 (59.8%) | 34 (53.9%) |
| GnRH-agonist long protocol | 59 (33.1%) | 103 (33.2%) | 97 (40.2%) | 29 (46.1%)[ab] |
| Basal FSH (mIU/ml) | 6.56 ± 2.05 | 6.53 ± 2.24 | 6.33 ± 1.75 | 6.01 ± 1.63 |
| Basal LH (mIU/ml) | 4.51 ± 1.85 | 5.13 ± 2.00[a] | 5.44 ± 2.05[a] | 5.29 ± 2.08[a] |
| Basal $E_2$ (pg/ml) | 33.47 ± 16.02 | 37.11 ± 15.11[a] | 38.75 ± 16.37[a] | 37.70 ± 14.94[a] |
| hCG day FSH (mIU/ml) | 13.53 ± 3.85 | 13.37 ± 3.96 | 13.60 ± 4.12 | 13.13 ± 3.83 |
| hCG day LH (mIU/ml) | 2.52 ± 1.86 | 3.00 ± 2.16[a] | 2.99 ± 2.08[a] | 3.07 ± 1.93 |
| hCG day $E_2$ (pg/ml) | 1520.91 ± 359.46 | 2494.17 ± 292.65[a] | 3484.82 ± 270.79[ab] | 4358.72 ± 268.01[abc] |
| hCG day P (ng/ml) | 0.55 ± 0.27 | 0.69 ± 0.29[a] | 0.77 ± 0.27[ab] | 0.83 ± 0.28[ab] |
| Endometrial thickness (mm), median (Q1-Q3) | 10.80 (9.60–11.80) | 10.90 (9.60–12.50) | 11.10 (9.50–12.55) | 10.70 (9.45–12.00) |
| No. of retrieved oocytes, median (Q1-Q3) | 10.0 (7.0–12.0) | 12.0 (10.0–15.0[a]) | 14.0 (11.0–17.0)[ab] | 14.0 (12.0–17.5)[ab] |
| No. of inviable embryos, median (Q1-Q3) | 2.00(1.00–3.00) | 2.00 (2.00–2.00) | 2.00 (2.00–2.00) | 2.00 (1.00–2.00) |
| No. of embryos transferred, n (%) | | | | |
| 1 | 167 (93.8%) | 278 (89.7%) | 223 (92.5%) | 54 (85.7%) |
| 2 | 11 (6.2%) | 32 (10.3%) | 18 (7.5%) | 9 (14.3%) |

**Notes.**

Data are presented as mean ± standard deviation unless otherwise indicated.

[a]Compared with set 1, $P < 0.05$.

[b]Compared with set 2, $P < 0.05$.

[c]Compared with set 3, $P < 0.05$.

Abbreviations: $E_2$, estradiol; Q, quartile; BMI, body mass index; IVF, in-vitro fertilization; ICSI, intracytoplasmic sperm injection; FSH, follicle stimulating hormone; LH, luteinizing hormone; hCG, human chorionic gonadotropin; P, progesterone.

both embryo stage groups. Compared to set 1 ($E_2 \leq 2,000$ pg/ml), higher $E_2$ levels in the cleavage group appeared to have a positive relationship with CPR, with the crude ORs of 1.54 (95% CI [1.27−1.29], $P < 0.001$), 1.64 (95% CI [1.28−2.10], $P < 0.001$) and 1.23 (95% CI [0.83−1.84], $P = 0.303$) from set 2 to set 4. As opposed to the cleavage group,

Meng et al. (2023), *PeerJ*, DOI 10.7717/peerj.15709

**Table 3  Pregnancy and neonatal outcomes according to different E$_2$ levels on hCG trigger day.**

| | Cleavage group | | | | | Blastocyst group | | | | |
|---|---|---|---|---|---|---|---|---|---|---|
| | Set 1 ($n = 1208$) | Set 2 ($n = 584$) | Set 3 ($n = 320$) | Set 4 ($n = 105$) | *P*-value | Set 1 ($n = 178$) | Set 2 ($n = 310$) | Set 3 ($n = 241$) | Set 4 ($n = 63$) | *P*-value |
| CPRs | 46.2% (558/1208) | 57.1% (333/584)[a] | 58.4% (187/320)[a] | 51.4% (54/105) | <0.001 | 56.2% (100/178) | 48.7% (151/310) | 54.4% (131/241) | 38.1% (24/63)[a] | 0.016 |
| Miscarriage rate | 17.7% (99/558) | 13.8% (46/333) | 14.9% (28/187) | 7.4% (4/54) | 0.085 | 18.0% (18/100) | 14.5% (22/151) | 13.7% (18/131) | 0 | 0.052 |
| Ectopic pregnancy rate | 2.2% (12/558) | 1.8% (6/333) | 1.6% (3/187) | 1.8% (1/54) | 0.745 | 0 | 0.6% (1/151) | 2.2% (3/131) | 0 | 0.231 |
| Live birth rate | 37.1% (448/1208) | 48.1% (281/584)[a] | 48.7% (156/320)[a] | 46.7% (49/105)[a] | 0.002 | 46% (82/178) | 41.3% (128/310) | 45.6% (110/241) | 34.9% (22/63) | 0.175 |
| singleton live birth rate | 27.4% (331/1208) | 32.2% (187/584)[a] | 35.3% (113/320)[a] | 31.4% (33/105) | 0.035 | 42.6% (76/178) | 35.5% (110/310) | 44.0% (106/241) | 34.9% (22/63) | 0.097 |
| Preterm birth rate | 9.0% (30/331) | 6.4% (12/187) | 5.3% (6/113) | 15.2% (5/33) | 0.212 | 13.1% (10/76) | 10.9% (12/110) | 13.2% (14/106) | 13.6% (3/22) | 0.908 |
| fetal weight | 3137 ± 498.4 | 3120 ± 482.7 | 3122 ± 465.2 | 3042 ± 498.8 | 0.760 | 3184 ± 455.0 | 3084 ± 495.1 | 3114 ± 495.5 | 3082 ± 519.4 | 0.206 |
| LBW rate | 7.6% (25/331) | 8.0% (15/187) | 6.2% (15/113) | 15.2% (5/33) | 0.375 | 5.2% (4/76) | 4.5% (5/110) | 5.7% (6/106) | 9.1% (2/22) | 0.580 |
| SGA rate | 6.9% (23/331) | 9.6% (18/187) | 14.2% (16/113) | 6.1% (2/33) | 0.111 | 5.3% (4/76) | 12.7% (14/110) | 12.3% (13/106) | 13.6% (3/22) | 0.285 |

**Notes.**

Data are presented as mean ± SD or n (%).

[a]Compared with set 1, $P < 0.05$.

Abbreviations: E$_2$, estradiol; CRP, clinical pregnancy rates; LBW, low birthweight; SGA, small for gestational age.

**Table 4  Univariate logistic analysis of clinical pregnancy rates for infertile patients from two embryo groups.**

| Variables | Cleavage group | | | Blastocyst group | | |
|---|---|---|---|---|---|---|
| | OR | 95% CI | P-value | OR | 95% CI | P-value |
| Age (years) | 0.92 | 0.91–0.94 | <0.001 | 0.96 | 0.93–0.98 | 0.016 |
| BMI (kg/m$^2$) | 0.99 | 0.96–1.02 | 0.715 | 0.98 | 0.94–1.03 | 0.499 |
| Duration of infertility (years) | 0.97 | 0.94–0.99 | 0.024 | 0.95 | 0.90–1.01 | 0.142 |
| Fertilization method (IVF *vs.* ICSI) | 1.02 | 0.72–1.33 | 0.774 | 1.14 | 0.94–1.71 | 0.121 |
| Previous cycle attempts | | | | | | |
| 1-2 *vs.* 0 | 0.62 | 0.50–0.77 | <0.001 | 1.02 | 0.67–1.56 | 0.932 |
| ≥3 *vs.* 0 | 0.12 | 0.06–0.23 | <0.001 | 0.59 | 0.19–1.83 | 0.387 |
| Sterility classification (Secondary *vs.* Primary) | 0.83 | 0.70–0.98 | 0.039 | 1.16 | 0.86–1.56 | 0.325 |
| Basal FSH (mIU/ml) | 0.92 | 0.89–0.95 | <0.001 | 0.98 | 0.91–1.05 | 0.580 |
| Basal LH (mIU/ml) | 1.02 | 0.97–1.06 | 0.327 | 0.94 | 0.87–1.01 | 0.120 |
| Basal E$_2$ (pg/ml) | 0.99 | 0.99–1.01 | 0.232 | 0.99 | 0.99–1.01 | 0.887 |
| hCG day FSH (mIU/ml) | 0.95 | 0.94–0.97 | <0.001 | 0.96 | 0.93–1.01 | 0.031 |
| hCG day LH (mIU/ml) | 0.98 | 0.94–1.01 | 0.179 | 1.03 | 0.94–1.11 | 0.527 |
| hCG day E$_2$ (pg/ml) | | | | | | |
| Set 1 | Reference | | | Reference | | |
| Set 2 | 1.54 | 1.27–1.89 | <0.001 | 0.75 | 0.52–1.18 | 0.124 |
| Set 3 | 1.64 | 1.28–2.10 | <0.001 | 0.97 | 0.66–1.43 | 0.869 |
| Set 4 | 1.23 | 0.83–1.84 | 0.303 | 0.52 | 0.29–0.94 | 0.0308 |
| hCG day P (ng/ml) | 0.95 | 0.73–1.24 | 0.708 | 0.50 | 0.31–0.80 | 0.004 |
| Endometrial thickness (mm) | 1.14 | 1.09–1.18 | <0.001 | 1.010 | 0.94–1.09 | 0.636 |
| No. of retrieved oocytes | 1.05 | 1.03–1.07 | <0.001 | 0.98 | 0.95–1.02 | 0.457 |
| No. of inviable embryos | 1.08 | 1.04–1.13 | 0.003 | 1.03 | 0.93–1.06 | 0.968 |
| No. of embryos transferred (2 *vs.* 1) | 2.30 | 1.80–2.93 | <0.001 | 1.76 | 1.05–2.93 | 0.031 |

**Notes.**

Abbreviations: E$_2$, estradiol; BMI, body mass index; IVF, in-vitro fertilization; ICSI, intracytoplasmic sperm injection; FSH, follicle stimulating hormone; LH, luteinizing hormone; hCG, human chorionic gonadotropin; P, progesterone; OR, odds ratio; CI, confidence interval.

higher E$_2$ in the blastocyst group was a negative factor for CPR, with crude ORs of 0.75 (95% CI [0.52−1.18], $P = 0.124$), 0.97 (95% CI [0.66−1.43], $P = 0.869$), 0.52 (95% CI [0.31−0.08], $P = 0.004$) from set 2 to set 4.

Table 5 lists the multivariate regression analysis results of both groups after adjusting for the confounding covariates. The constructed model I adjusted for demographic factors (including age and BMI), and model II adjusted for all covariates. In the cleavage group, the adjusted ORs of model I and model II for CPR with patients in set 4 (E$_2$ >4,000 pg/ml) were 0.98 and 1.59, respectively, compared to those in the reference set 1 (E$_2$ ≤ 2,000 pg/ml). The statistical difference was not significant ($P = 0.919$ in model I, $P = 0.294$ in model II). The E$_2$ level on hCG trigger day in set 4 of the blastocyte group was an independent risk factor for CPR, with an adjusted OR of 0.50 (95% CI [0.28−0.90], $P = 0.021$) in model I and 0.43 (95% CI [0.24−0.97], $P = 0.039$) in model II.

Meng et al. (2023), *PeerJ*, DOI 10.7717/peerj.15709

**Table 5   Multivariate logistic model for clinical pregnancy rates of two embryo stage groups.**

| E2 (pg/ml) on the trigger day | Cleavage group | | | | Blastocyst group | | | |
|---|---|---|---|---|---|---|---|---|
| | Model I | | Model II | | Model I | | Model II | |
| | OR (95% CI) | *P*-value | OR (95% CI) | *P*-value | OR (95% CI) | *P*-value | OR (95% CI) | *P*-value |
| Set 1 (As reference) | 1.0 | – | 1.0 | – | 1.0 | – | 1.0 | – |
| Set 2 | 1.29 (1.04, 1.58) | 0.018 | 1.37 (0.99, 1.87) | 0.053 | 0.73 (0.50, 1.06) | 0.101 | 0.78(0.506, 1.186) | 0.241 |
| Set 3 | 1.36 (1.05, 1.76) | 0.019 | 1.48 (0.98, 2.24) | 0.065 | 0.95 (0.64, 1.41) | 0.804 | 1.07(0.67, 1.72) | 0.774 |
| Set 4 | 0.98 (0.65, 1.48) | 0.919 | 1.59 (0.77, 3.30) | 0.294 | 0.50 (0.28, 0.90) | 0.021 | 0.43(0.24, 0.97) | 0.039 |

**Notes.**

Model I: adjusted for age (smooth) and BMI.

Model II: The following confounders were adjusted: age (smooth), BMI, infertility duration, fertilization method, sterility classification, previous cycle attempts, basal FSH, basal LH, basal $E_2$, FSH on hCG day, LH on hCG day, P on hCG day, endometrial thickness on hCG day, number of retrieved oocytes, the number of inviable embryos, and the number of embryos transferred.

Abbreviations: OR, odds ratios; CI, Confidence interval.

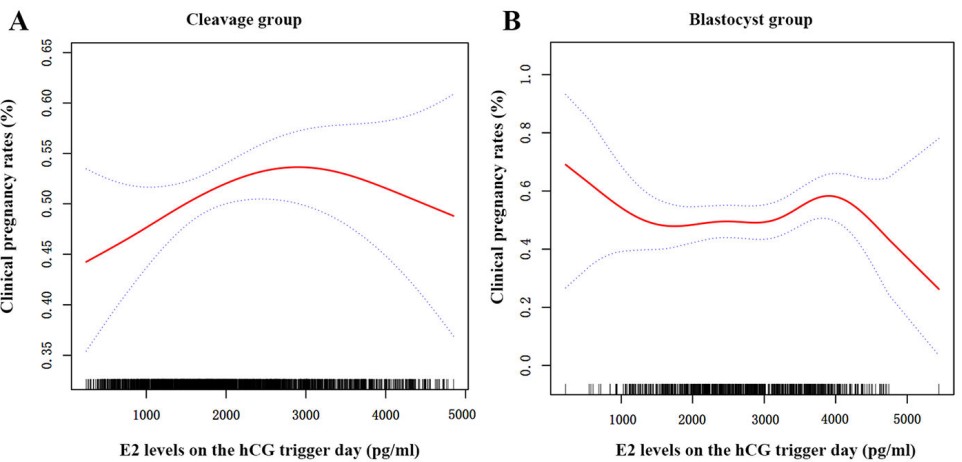

**Figure 2** The curvilinear relationship between CPR and $E_2$ levels on hCG trigger day in two groups. A nonlinear association between serum $E_2$ levels on hCG trigger day and CPR of patients in the cleavage group (A) and the blastocyst group (B). The red lines represent the smoothing curve fit between serum $E_2$ and CPR, and the blue lines represent the 95% confidence interval.

## Non-linear relationship between serum $E_2$ and clinical pregnancy rates

After adjusting for all demographic and clinical variables, we found a nonlinear relationship between the $E_2$ levels on trigger day and the CPR for the two groups using spline smoothing plots (Fig. 2). The threshold effects of serum $E_2$ on CPR were analyzed using the two-piecewise linear regression analysis (Table 6). Inflection points for the cleavage and blastocyst groups were calculated as 31.1 pg/dl and 39.7 pg/dl (1 pg/dl =100 pg/ml), respectively. In the cleavage group, the CPR was positively associated with serum $E_2$ when the serum $E_2$ was under 31.1 pg/dl (adjusted OR = 1.04, 95% CI [1.01−1.07], $P = 0.022$); however, when the $E_2$ levels were beyond 31.1 pg/dl, the correlation between $E_2$ and CPR was not statistical significantly (adjusted OR = 0.98, 95% CI [0.96−1.01], $P = 0.297$). In the blastocyst group, the CPR did not change significantly when $E_2$ levels were lower than 39.7 pg/dl (adjusted OR = 0.99, 95% CI [0.97−1.02], $P = 0.846$). However, when the $E_2$ levels were higher than 39.7 pg/dl, the CPR decreased by 17% with every 1 pg/dl increase in $E_2$ (adjusted OR = 0.83, 95% CI [0.72–0.96], $P = 0.012$).

## DISCUSSION

Supraphysiologic $E_2$ conditions are unavoidable in fresh ET cycles after ovarian stimulation, and the influence of such high $E_2$ concentrations on clinical outcomes remain controversial. Our findings showed that the detrimental impact of elevated serum $E_2$ on CPR was not the same between cleavage-stage and blastocyst ET. Infertile women with hCG $E_2$ >4,000 pg/ml who received a fresh blastocyst transfer had significantly lower CPR than those with hCG $E_2 \leq 2,000$ pg/ml. Additionally, the inflection points for the cleavage and blastocyst groups were 31.1 pg/dl and 39.7 pg/dl, respectively. The threshold effects showed that CPR of patients in cleavage group gradually increased until serum E2 reached 31.1 pg/dl and then remained at a particular level, although the total change was not significant. As for the

**Table 6   The results of two-piecewise linear regression model.**

| Groups | Inflection point (hCG $E_2$, pg/dl) | Adjusted OR (95% CI) | $P$-value | LRT test |
|---|---|---|---|---|
| Cleavage group | <31.1 | 1.04 (1.01–1.07) | 0.022 | 0.029 |
|  | >31.1 | 0.98 (0.96–1.01) | 0.297 |  |
| Blastocyst group | <39.7 | 0.99 (0.97–1.02) | 0.846 | 0.013 |
|  | >39.7 | 0.83 (0.72–0.96) | 0.012 |  |

**Notes.**

Adjustment covariates: age (smooth), BMI, infertility duration, fertilization method, sterility classification, previous cycle attempts, basal FSH, basal LH, basal $E_2$, FSH on hcg day, LH on hCG day, P on hCG day, endometrial thickness on hCG day, number of retrieved oocytes, the number of inviable embryos, and the number of embryos transferred.

$P < 0.05$ presented a nonlinear relationship.

Abbreviations: LRT test, logarithmic likelihood ratio test; OR, odds ratios; CI, Confidence interval.

blastocyst group, when $E_2$ level was over 39.7 pg/dl, the CPR decreased by about 17% with every 1 pg/dl increase in serum $E_2$. However, there was no significant correlation between serum $E_2$ and CPR when $E_2$ level was lower than 39.7 pg/dl.

The serum $E_2$ on the hCG trigger day expressed a concentration-dependent effect on pregnancy outcomes. *Joo et al. (2010)* reported that the implantation rate increased until the serum $E_2$ equaled 4,000 pg/ml. Similarly, another retrospective study of 3,033 fresh cycles indicated that $E_2$ levels of at least 4,000 pg/ml on trigger day had an increased rate of small gestational age (adjusted OR = 1.65, 95% CI [1.05–2.59]) compared to the those with $E_2$ <2,000 pg/ml (*Zhang, Du & Sun, 2022*). Our data further supported the adverse effect of a supraphysiological $E_2$ environment on endometrial receptivity. Interestingly, we found that the adverse impact of increased $E_2$ on CPR regarding the embryo stage (cleavage stage and blastocyst stage) in fresh cycles was different. According to a prior multicenter, non-blinded RCT comparing frozen *versus* fresh blastocyst-stage ET, higher live birth rates were observed in the frozen-thawed transfer cycle than in the fresh cycle (*Wei et al., 2019*). Meanwhile, no significant difference was seen in the rates of implantation, pregnancy, pregnancy loss, or live birth in two sizable studies in cleavage-stage ET using the same experimental methodology (*Shi et al., 2018*; *Vuong et al., 2018*). It remained unclear what caused the differences in results between cleavage-stage ET and blastocyst-stage ET. One of the mechanisms may be the supraphysiological condition after COS, which advanced the window of implantation, thereby decreasing endometrial receptivity for fifth days after ovum pick-up (*Basir et al., 2001*). The induction of a partial implantation window that closed had a direct impact on blastocyst implantation but had little or no impact on cleavage-stage embryo (*Wei et al., 2019*). Therefore, our results suggested that supraphysiological $E_2$ concentrations would have a greater impact on the CPR of fresh blastocyst ET.

*Ying et al. (2021)* suggested that an elevated $E_2$ level on hCG trigger day may contribute to the incidence of OHSS. Increased capillary permeability and arteriolar vasodilatation were the primary pathophysiological alterations of OHSS (*Balen et al., 2016*). It is unknown whether elevated serum $E_2$ will affect reproductive outcomes through the pathophysiological state of OHSS. A previous retrospective cohort experiment compared the pregnancy outcomes of OHSS ($N = 190$) and non-OHSS ($N = 197$) patients who

underwent fresh ET cycles. The findings demonstrated that there was no obviously adverse effect of OHSS on the subsequent results (*Balen et al., 2016*). To investigate the effects of late-onset OHSS on obstetric outcomes, another prospective observational study included 17,537 patients receiving fresh ET and found that the live birth rate did not significantly differ between the OHSS and matched control groups (*Hu et al., 2021*). Consequently, the prevalence of OHSS was not taken into account as a confounding factor in our study.

The plausible biological mechanisms underlying high $E_2$ levels affecting CPR in fresh cycles remain unclear. According to previous research, high $E_2$ levels could intensify uterine contractility, weaken the endometrial blood flow, and alter endometrial gene expression profiles when serum $E_2$ reaches a certain threshold (*De Ziegler et al., 1998*; *Haouzi et al., 2009*; *Ng et al., 2006*). *Chou et al. (2020)* reported that supraphysiological $E_2$ concentrations stimulated endometrial epithelial cell apoptosis, increased extramitochondrial reactive oxygen species production, decreased ATP formation, and resulted in mitochondrial dysfunction. Furthermore, high $E_2$ concentrations may exert toxins directly on embryos which could be deleterious to embryo adhesion in vitro (*Valbuena et al., 2001*). *Chang et al. (2022)* found that blastocyst proliferation was reduced, and apoptotic cells increased following exposure to a high $E_2$ culture environment. They also confirmed the direct impact of high $E_2$ concentrations on implantation and early post-implantation development on blastocysts within the ET mice model. Additionally, *Kolibianakis et al. (2002)* observed the early endometrial maturation in fresh cycles using aspirational endometrial biopsies, indicating an advanced implantation window in fresh ET. Given this evidence from both animals and humans, elevated $E_2$ levels during ovarian stimulation might negatively affect endometrial receptivity and embryo quality, thus disturbing clinical pregnancy.

This study's strengths involve its large sample size ($n = 3,009$), which was larger than most similar research after minimizing the selection and statistical bias. To account for the embryo stage affecting pregnancy outcomes, this study separately analyzed the cleavage-stage subgroup *versus* the blastocyst subgroup in fresh ET cycles. Moreover, our findings indicated that when $E_2$ levels >39.7 pg/dl on hCG trigger day, the fresh blastocyst ET should be suspended. This practice could be applied in the clinical setting.

There are some limitations of this study. First, our study was built on a retrospective cohort. Recall bias and inconsistencies regarding the medication history of patients were inevitable. Second, this study was a single-center-based design. The inherent weakness of a single center is apparent, as the results may not apply to other institutions. Moreover, the mechanism underlying the discrepant outcomes between fresh cleavage-stage ET and blastocyst ET under the supraphysiological $E_2$ condition is also unknown. In order to further increase its generalizability, a larger prospective multicenter study and robustly designed basic experiment will be conducted.

## CONCLUSION

The results of our study demonstrated that elevated $E_2$ levels on hCG trigger day might negatively affect the CPR of patients in the fresh blastocyst ET. However, it has no similar effect on the CPR of patients in the fresh cleavage-stage ET. We should adopt an appropriate

strategy in fresh cycles if excessive $E_2$ concentration appears in daily clinical practice. More prospective studies are needed to elucidate this underlying mechanism.

### Funding

This study was supported by the Special Correspondent Project of Guangdong Rural Science, and Technology (No. KTPYJ2021015), the Third Affiliated Hospital of Sun Yat-Sen University, Clinical Research Program (No. YHJH202209), and a grant from the MOE Key Laboratory of Gene Function and Regulation. The funders had no role in study design, data collection and analysis, decision to publish, or preparation of the manuscript.

### Grant Disclosures

The following grant information was disclosed by the authors:
Special Correspondent Project of Guangdong Rural Science, and Technology: KTPYJ2021015.
The Third Affiliated Hospital of Sun Yat-Sen University, Clinical Research Program: YHJH202209.
MOE Key Laboratory of Gene Function and Regulation.

### Competing Interests

The authors declare there are no competing interests.

### Author Contributions

- Yue Meng conceived and designed the experiments, performed the experiments, analyzed the data, prepared figures and/or tables, authored or reviewed drafts of the article, and approved the final draft.
- Linlin Tao performed the experiments, authored or reviewed drafts of the article, and approved the final draft.
- Tingting Xia analyzed the data, prepared figures and/or tables, and approved the final draft.
- Jieru Zhu analyzed the data, prepared figures and/or tables, and approved the final draft.
- Xiaoqi Lin performed the experiments, prepared figures and/or tables, and approved the final draft.
- Wen Zhou analyzed the data, prepared figures and/or tables, and approved the final draft.
- Yuxia Liu analyzed the data, prepared figures and/or tables, and approved the final draft.
- Jianping Ou conceived and designed the experiments, prepared figures and/or tables, authored or reviewed drafts of the article, and approved the final draft.
- Weijie Xing conceived and designed the experiments, authored or reviewed drafts of the article, and approved the final draft.

### Human Ethics

The following information was supplied relating to ethical approvals (i.e., approving body and any reference numbers):

This study has been approved by the Ethics Committee of the Third Affiliated Hospital of Sun Yat-Sen University (II2023-051-01).

## Data Availability

The raw data are available in the Supplemental Files.

## Supplemental Information

Supplemental information for this article can be found online at http://dx.doi.org/10.7717/peerj.15709#supplemental-information.

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
