# Peer review of "Elevated estradiol levels on hCG trigger day adversely effects on the clinical pregnancy rates of blastocyst embryo transfer but not cleavage-stage embryo transfer in fresh cycles: a retrospective cohort study"

_PeerJ, doi:10.7717/peerj.15709_

## Round 0.1 · original submission · Major Revisions

Editor notes and agrees with the review’s concerns. Additional concerns and comments are listed below;

Please clarify if the experimental design of this study includes mixed data from embryos that originated from IVF or ICSI or both. It is not clear in the materials and methods.

The embryology protocols are missing such as the insemination method, culture conditions, media used, fertilization rate, embryo grades, etc.

What was the reason for cleavage stage embryo transfers? Please explain this in the limitations.

Male factor is neglected in this study. There are not any sperm parameters mentioned. Were testicular sperm or ejaculated sperm used for the procedures?
What are the sperm concentration, motility, and morphology of males within groups?

English editing is needed.

This manuscript should be revised extensively and the statistical analysis must be redone by removing the biased data before resubmission.

Best Regards

Reviewer 1 ·

Basic reporting

no comment

Experimental design

E2 over 3354pg/ml or 3500 pg/ml were believed to highly indicated of OSHH, but the article did not mention possible pathophysiological impact of OSHH on pregnancy, and possible lead to low CPR of set 4 vs. set 1.
Even though severe OSHH was excluded, the author did not clearly state the criteria of exclusion.

Validity of the findings

In the discussion part, did not discuss thoroughly the reason of set 4 of cleavage stage CPR was not impacted by high E2 level.

Additional comments

no comments

Reviewer 2 ·

Basic reporting

The manuscript is prepared with good structure. However an English editor can improve English structure and grammar of the manuscript

Experimental design

The downside is that the study is retrospective review. Its is doubtful that in the present clinical scenario the study has much relevance.
The authors need to provide the reasons and evidence for the relevance of the study.

Validity of the findings

Findings have been provided with good statistical analysis. Why the high Estradiol levels had no deleterious effects on early cleavage stage embryos?

Additional comments

The study design was not robust as it was a retrospective study.
The authors need to explain and provide reasons for higher CPR obtained with the early fresh cleavage stage transfers vs blastocyst transfer. Also the authors can provide the evidence of any similar studies published in literature supportive of their findings/
Please provide conclusions on the clinical impact of the paper and whether it changes any practice guidelines.
Please explain in the current scenario how does the study impact IVF practice when most IVF centers are performing frozen blastocysts transfers.

Annotated reviews are not available for download in order to protect the identity of reviewers who chose to remain anonymous.

---

## Round 0.2 · accepted · Accept

This article is acceptable for publication but please consider the minor revisions suggested by the reviewer.

Reviewer 2 ·

Basic reporting

The manuscript has undergone major revision. The authors have provided robust responses and improved the quality of the manuscript.

Experimental design

Good retrospective study with large patient population.

Validity of the findings

It is a well written study with good statistical analysis and the authors have explained the study result and the impact of the elevated estradiol on various embryonic stage transfers.

Additional comments

I have attached the PDF with few minor changes in the manuscript.

Annotated reviews are not available for download in order to protect the identity of reviewers who chose to remain anonymous.